# Transcriptome analysis of two isolates of the tomato pathogen *Cladosporium fulvum*, uncovers genome-wide patterns of alternative splicing during a host infection cycle

**Alex Z. Zaccaron[1,2], Li-Hung Chen[1¤], Ioannis Stergiopoulos[1]** *

**1** Department of Plant Pathology, University of California Davis (UC Davis), Davis, California United States of America, **2** Integrative Genetics and Genomics Graduate Group, University of California Davis (UC Davis), California, United States of America

¤ Current address: National Chung Hsing University (NCHU), Taichung, Taiwan.
* istergiopoulos@ucdavis.edu

**Data Availability Statement:** The raw RNAseq reads were deposited in the NBCI's sequence reads archive (SRA) under accessions SRR29437234-

## Abstract

Alternative splicing (AS) is a key element of eukaryotic gene expression that increases transcript and proteome diversity in cells, thereby altering their responses to external stimuli and stresses. While AS has been intensively researched in plants and animals, its frequency, conservation, and putative impact on virulence, are relatively still understudied in plant pathogenic fungi. Here, we profiled the AS events occurring in genes of *Cladosporium fulvum* isolates Race 5 and Race 4, during nearly a complete compatible infection cycle on their tomato host. Our studies revealed extensive heterogeneity in the transcript isoforms assembled from different isolates, infections, and infection timepoints, as over 80% of the transcript isoforms were singletons that were detected in only a single sample. Despite that, nearly 40% of the protein-coding genes in each isolate were predicted to be recurrently AS across the disparate infection timepoints, infections, and the two isolates. Of these, 37.5% were common to both isolates and 59% resulted in multiple protein isoforms, thereby putatively increasing proteome diversity in the pathogen by 31% during infections. An enrichment analysis showed that AS mostly affected genes likely to be involved in the transport of nutrients, regulation of gene expression, and monooxygenase activity, suggesting a role for AS in finetuning adaptation of *C. fulvum* on its tomato host during infections. Tracing the location of the AS genes on the fungal chromosomes showed that they were mostly located in repeat-rich regions of the core chromosomes, indicating a causal connection between gene location on the genome and propensity to AS. Finally, multiple cases of differential isoform usage in AS genes of *C. fulvum* were identified, suggesting that modulation of AS at different infection stages may be another way by which pathogens refine infections on their hosts.

SRR29437254 for isolate Race 5, and SRR29424125-SRR29424145 for isolate Race 4. The assembled transcripts and their expression values were deposited in a public repository, available at https://zenodo.org/records/11176736.

**Funding:** This work was supported by the National Science Foundation Division of Integrative Organismal Systems (NSF-IOS) Award number 1557995 to IS. The funding agency had no role in study design, data collection and interpretation, or the decision to submit the work for publication.

## Author summary

Alternative splicing (AS) is a major source of transcriptome plasticity, proteome diversity, and phenotypic complexity in eukaryotes. Here, we analyzed the AS events happening in two isolates of the tomato pathogen *Cladosporium fulvum*, when infecting their host. We reveal an extensive infection-to-infection and isolate-to-isolate variation in the transcript isoforms assembled from transcribed pathogen genes, indicating that species-level inferences on AS cannot be reliably made based on single infections or isolates. Nonetheless, we found that AS is prevalent in pathogen genes and likely has a multifactorial effect on host infections, as a core set of genes that mostly encode transporters, transcription factors, and cytochrome P450 enzymes are recurrently AS in *C. fulvum* during infections. We finally show that genes in repeat-rich regions of the genome are more frequently affected by AS and that the pathogen may prime infections by the selective up- or downregulation of specific isoforms at different infection stages.

## Introduction

Alternative splicing (AS) is a molecular process by which diverse mature mRNA molecules are produced by single genes, as a result of aberrant intron splicing during transcription [1]. There are five main types of AS events, including exon skipping (ES), mutually exclusive exons (MX), intron retention (IR), alternative 5' or 3' splice sites (A5/A3), and alternative first or last exons (AF/AL) [2]. A key outcome of AS is that it increases mRNA and protein diversity in cells, but it may further affect several other aspects of mRNA metabolism, thereby modulating gene expression at the post-transcriptional level [1]. Aberrant mRNA splicing, for instance, often results in isoforms that are the targets for nonsense-mediated mRNA decay (NMD), an mRNA surveillance system that detects and degrades prematurely terminated transcripts generated by errors in mRNA processing. AS variants can also regulate the abundance of functional transcripts via a mechanism known as regulated unproductive splicing and translation (RUST). It is thus not surprising that a strong correlation exists between AS and functional complexity in cells that affects many of their cellular, metabolic, and physiological processes, including responses to environmental stressors and other stimuli [3–5]. In humans, for instance, it is estimated that 95% of all genes undergo AS at different developmental stages of tissues [6,7], thereby often leading to hereditary diseases and cancers. Likewise, in plants, AS has been shown to have a significant functional impact in processes such as photosynthesis, flowering, abiotic stress responses, and defense against microbial attacks [8–17].

While the biological significance of AS in plants and mammals is well documented, AS in fungi is in contrast less investigated. Studies have shown that the frequency of AS varies considerably among fungal species and it is typically low compared to other eukaryotes, ranging from less than 1% in the baker's yeast *Saccharomyces cerevisiae* to 48.9% of the genes in the biocontrol fungus *Trichoderma longibrachiatum* [18–21]. Despite such variations, evidence suggests that AS significantly impacts a plethora of cellular and physiological processes in fungi, including growth and development [22], response to environmental stressors [23–25], histone deubiquitination [26], gene expression [27,28], secretion of enzymes [29], subcellular localization of proteins [30,31], and others [18,19]. An increasing body of evidence also indicates that AS may affect virulence of pathogenic fungi and resistance to antifungal compounds [18–20,32–34]. For instance, in *Sclerotinia sclerotiorum* that causes white mold disease on an array of plant species, the landscape of AS events changes in accordance to the host plant that the pathogen is infecting [35], suggesting that it is important to host adaption. In the rice blast

fungus *Magnaporthe oryzae*, the frequency of AS appears to increase during host infections, with many AS events putatively being upregulated [33]. A genome-wide analysis of AS in human pathogenic fungi also showed that AS is more frequent during stress and likely to play important regulatory roles during infection [32]. Collectively, such studies highlight that AS could be promoting virulence and adaptation of fungi to their hosts and environment.

*Cladosporium fulvum* (syn. *Passalora fulva*, *Fulvia fulva*) is a hemibiotrophic fungal pathogen that causes tomato leaf mold [36]. Over the last 40 years, this fungus has been a valuable model for the study of plant-microbe interactions [37,38]. Recently, high-quality chromosome-level genome assemblies were obtained for *C. fulvum* isolates Race 5 and Race 4 [39,40], thereby laying the ground for more in depth genomic and transcriptomic studies in this pathogen. Ensuing analyses revealed that the genome of *C. fulvum* is organized into 13 core and 2 dispensable chromosomes, is ~39% repetitive, and it is arranged in a 'checkerboard' pattern of gene-rich/repeat-poor regions, interspersed with gene-poor/repeat-rich regions, in accordance with the 'two-speed genome' model of evolution [41]. Although significant progress has been made in understanding the genomic architecture of *C. fulvum* [39,40], its transcriptome and particularly profile of AS during infections remained largely unexplored.

In this study, we sequenced the transcriptomes of isolates Race 5 and Race 4, and analyzed the landscape of AS events occurring in pathogen genes nearly throughout the infection process and in three independent tomato infections. This was done in order to assess the reproducibility of AS events and, based on their conservation across different isolates, infections, and infection stages, identify the ones that are most likely to be functionally relevant for virulence rather than being splicing noise. Our studies revealed a high frequency and dynamic spectrum of AS events taking place in the two isolates during host infections that is particularly affecting genes encoding major facilitator superfamily transporters, sugar transporters, transcription factors, and cytochrome P450 enzymes, but less candidate effectors, thereby suggesting a role for AS in adaptation of *C. fulvum* on its tomato host during infections.

## Results

### Transcriptome profiling of *C. fulvum* during compatible host infections reveals extensive transcript isoform heterogeneity among isolates and infections

The transcriptomes of *C. fulvum* isolates Race 5 and Race 4 were sequenced at high depth during their interaction with *Solanum lycopersicum* cv. Moneymaker at seven timepoints (2, 4, 6, 8, 10, 12, and 14 dpi) and from three independent infections (i.e. biological replicates) (Tables 1 and S1–S3 and S1 Text). The RNAseq reads obtained were first used to confirm that the previously predicted exon-intron structures in the intron-containing genes of isolates Race 5 and Race 4 [40] were well-supported by the data (S2 Text). An ensuing investigation into whether orthologous genes in isolates Race 5 and Race 4 had similar number and size of introns that further shared similar start and end coordinates, showed that from the 14,747 ortholog gene pairs, 14,739 (99.9%) pairs had the same number of introns in both orthologous genes (Fig 1A), 14,648 (99.3%) pairs had the same total size of intronic sequences (Fig 1B), and 14,729 (99.9%) had the same number of introns with the same start and end coordinates. This indicated a high conservation of gene structures (S2 Text), thereby enabling a comparative analysis of AS events in the two isolates.

The RNAseq reads were next used to perform reference-based transcriptome assemblies and to further obtain a set of unique transcript isoforms from isolates Race 5 and Race 4 (S1 Fig and S4 Table and S3 Text). A total of 57,148 unique transcript isoforms could be assembled for the two isolates, 43,824 of which originated from genes of isolate Race 5 and 41,173

**Table 1. Number and percentage of RNAseq reads that mapped to the genome of *Cladosporium fulvum*.** The table shows numbers and percentages of RNAseq reads obtained from three independent infections (i.e. biological replicates; Rep.) and seven timepoints during the interaction between the *C. fulvum* and tomato that mapped to the genomes of *C. fulvum* isolates Race 5 and Race 4.

| Isolate | Timepoint | Rep. 1<br>No. of reads (%) | Rep. 2<br>No. of reads (%) | Rep. 3<br>No. of reads (%) |
|---|---|---|---|---|
| Race 5 | 2 dpi | 135,953 (0.08%) | 150,535 (0.10%) | 195,887 (0.10%) |
| Race 5 | 4 dpi | 231,260 (0.16%) | 313,138 (0.18%) | 437,311 (0.20%) |
| Race 5 | 6 dpi | 3,270,510 (2.22%) | 3,036,751 (1.91%) | 5,898,146 (2.67%) |
| Race 5 | 8 dpi | 23,392,396 (12.08%) | 16,071,949 (8.96%) | 19,383,926 (8.95%) |
| Race 5 | 10 dpi | 22,863,783 (29.57%) | 60,588,111 (32.10%) | 17,654,682 (17.53%) |
| Race 5 | 12 dpi | 27,109,200 (32.55%) | 34,630,359 (36.42%) | 28,289,494 (34.16%) |
| Race 5 | 14 dpi | 27,894,580 (40.58%) | 30,191,286 (35.07%) | 37,633,905 (43.87%) |
| Race 4 | 2 dpi | 409,880 (0.23%) | 1,217,740 (0.60%) | 1,237,896 (0.62%) |
| Race 4 | 4 dpi | 736,856 (0.34%) | 976,981 (0.51%) | 1,852,149 (0.92%) |
| Race 4 | 6 dpi | 712,610 (0.45%) | 2,970,591 (1.55%) | 1,449,788 (0.95%) |
| Race 4 | 8 dpi | 2,013,657 (1.48%) | 17,251,769 (10.86%) | 2,229,050 (1.59%) |
| Race 4 | 10 dpi | 10,724,443 (13.49%) | 23,799,801 (26.02%) | 2,530,422 (2.56%) |
| Race 4 | 12 dpi | 23,492,901 (29.28%) | 25,559,655 (36.13%) | 12,719,704 (19.52%) |
| Race 4 | 14 dpi | 28,462,052 (35.52%) | 45,780,891 (57.64%) | 26,803,887 (38.11%) |

originated from genes of isolate Race 4. However, only 27,849 (49%) of them were common to both isolates, whereas 15,975 (28%) and 13,324 (23%) were unique to isolate Race 5 and Race 4, respectively (S2A Fig). This indicates that although similar numbers of unique transcript isoforms were generated from orthologous genes during infections, these varied substantially between the two isolates. A similar trend was also seen when examining the extent to which individual genes generated the same pool of transcript isoforms across different infections (i.e. biological replicates) (S2B and S2C Fig) or timepoints of the infection process (S3 Fig).

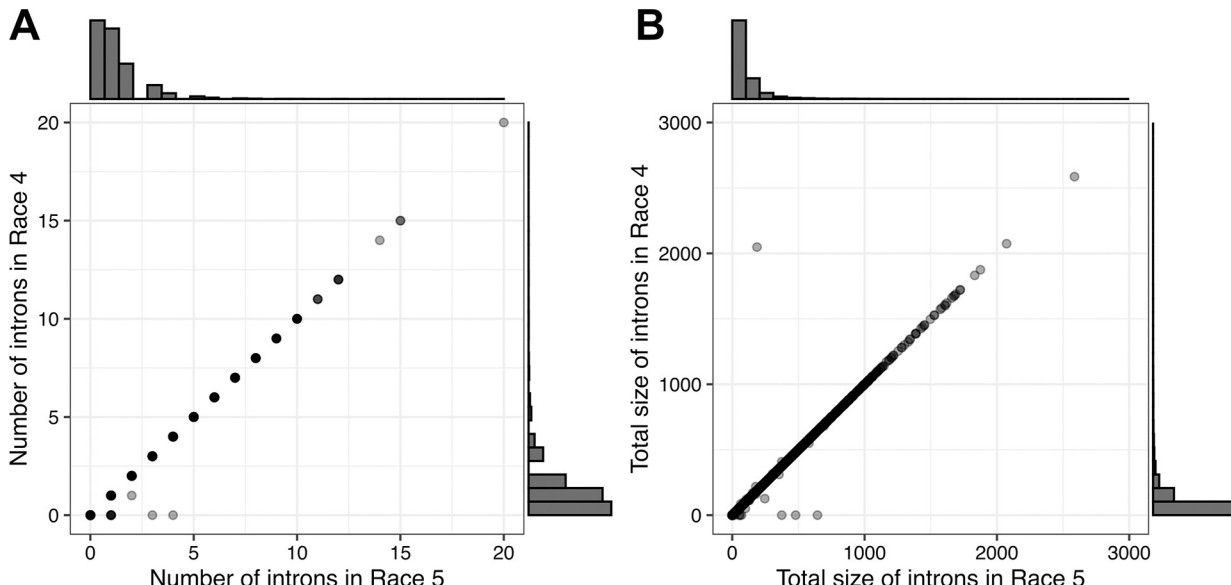

**Fig 1. Number and size of introns that are conserved among pairs of orthologous genes in *Cladosporium fulvum* isolates Race 5 and Race 4.** (A) Number of introns predicted in the orthologous genes of isolates Race 5 and Race 4. (B) Total size of introns in base pairs (bp) between orthologous genes. Both scatter plots were generated by comparing 14,747 one-to-one orthologous gene pairs from the genome annotations of isolates Race 5 [39] and Race 4 [40].

Collectively, these results indicate that extensive heterogeneity in transcript isoforms exists among biological samples, which could be biologically driven from stochastic fluctuations in gene transcription and splicing events and/or due to technical noise in the data.

Further analysis of the 57,148 transcript isoforms that were assembled from the two isolates showed that most were assembled in only one sample, i.e. in one infection replicate of one timepoint of the interaction of one isolate with the host. Such singleton transcript isoforms accounted for 12,688 (79.4%) of the 15,975 transcript isoforms assembled from isolate Race 5, and 11,325 (85.0%) of the 13,324 transcript isoforms assembled from isolate Race 4, and they likely represented spurious or random transcriptional events that were therefore removed from the analyses. As a result, the total number of unique transcript isoforms reduced to 33,135, which included 31,136 and 29,848 transcript isoforms from isolates Race 5 and Race 4, respectively (S4A Fig). As expected, the removal of the singleton transcript isoforms from the dataset increased the percentage of transcript isoforms shared by the two isolates increased from 49% to 84%, and had a similar effect on the percentages of transcript isoforms supported by all three infection replicates in each isolate (S4B Fig) and at each timepoint (S5 Fig). Collectively, these results indicate that most isoforms generated by pathogen genes during infections are singletons that likely represent splicing or transcriptional noise of potentially minimal functional significance.

## A core set of genes in *C. fulvum* are recurrently AS across different infections and isolates

To further increase the accuracy of our ensuing AS analysis, the 33,135 non-singleton transcript isoforms were filtered such that isoforms that shared all their intron coordinates (or lack of them) were considered as duplicates, and thus only the longest of them were kept. Following this filtering step, the total number of unique transcript isoforms was reduced to 33,559, including 26,818 transcript isoforms from isolate Race 5 and 26,397 from isolate Race 4. The transcript isoforms were then mapped to the genes of each isolate, and genes associated with more than one transcript isoform were now essentially considered as AS. This included 6,034 genes from isolate Race 5, or 40.3% of its total protein-coding genes, and 6,069 genes from isolate Race 4, or 40.5% of its total protein-coding genes (S5 Table). Among the AS genes, 5,611 genes, or 92.3% of the AS genes in isolate Race 5 and 92.5% of the AS genes in isolate Race 4, were common to both isolates, with 4,111 (73.3%) of them further generating the same number of transcript isoforms in each isolate (S6 Table). The percentage of AS genes shared between the two isolates is high, but it is likely to be somewhat inflated by the removal of the singleton transcript isoforms from the analysis. Even so, 37.4% (i.e. 5,611 of the 14,993 genes) and 37.7% (i.e. 5,611 of the 14,895 genes) of the protein-coding genes in isolates Race 5 and Race 4, respectively, were AS spliced and common to both isolates, suggesting that within the noise of the transcript heterogeneity that is created during infections, a sizeable core set of genes exists that are recurrently AS.

Inspection of the transcript diversity generated by the AS genes showed that most of the AS genes in isolates Race 5 ($n = 4,590$; 76.1%) and Race 4 ($n = 4,669$; 76.9%) produced two or three transcript isoforms, and only a small number of AS genes from isolates Race 5 ($n = 55$; 0.9%) and Race 4 ($n = 45$; 0.7%) produced larger numbers of 10 or more transcript isoforms (Fig 2A). An analysis of the AS events revealed similar patterns for isolates Race 5 and Race 4, with IR being the most frequent type of AS, and accounting for 71.1% and 70.7% of all classified AS events in isolates Race 5 and Race 4, respectively (Fig 2B and 2C and S7 Table). Collectively these results indicate similar patterns of AS events in the two isolates and that although many genes are predicted to be AS, they overall generate a low number of transcript isoforms.

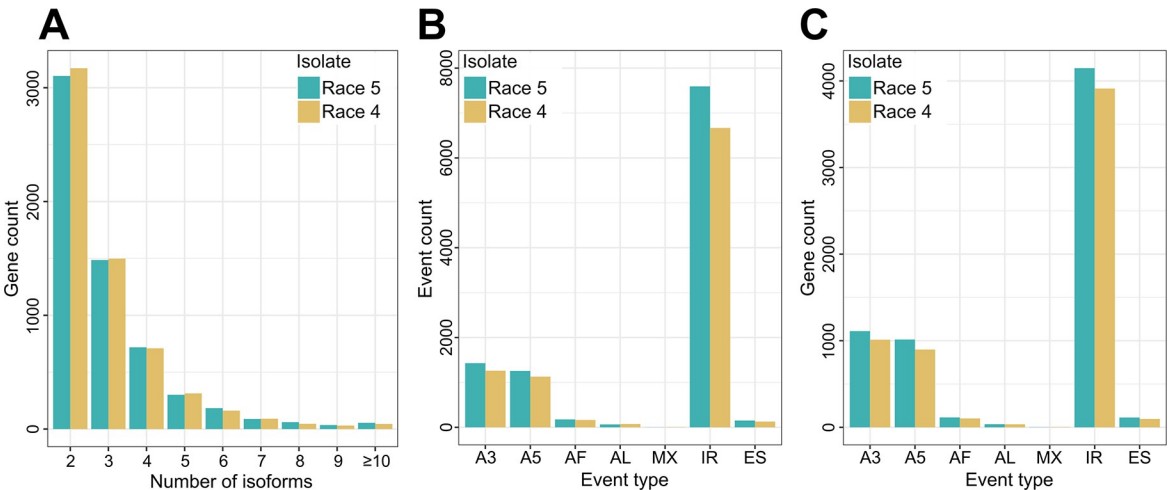

**Fig 2. Genes that are alternatively spliced (AS) in *Cladosporium fulvum* isolates Race 5 and Race 4 during tomato infections, generally produce low numbers of transcript isoforms and exhibit mostly intron retention events.** (A) Bar chart showing the number of AS genes from isolates Race 5 and Race 4 (y-axis), producing two or more transcript isoforms (x-axis). (B) Bar chart showing the number of AS events in isolates Race 5 and Race 4 (y-axis), classified in one of the major types of AS (x-axis). (C) Bar chart showing the number of AS genes in isolates Race 5 and Race 4 (y-axis), classified in one of the major types of AS (x-axis). In panels B and C, types of AS events shown are alternative 5' or 3' splice sites (A5/A3), alternative first or last exons (AF/AL), mutually exclusive exons (MX), intron retention (IR), and exon skipping (ES).

We next investigated whether AS in pathogen genes during host infection differentially affected different gene categories, by performing a functional enrichment analysis based on conserved PFAM domains, gene ontology (GO) terms, and functional gene categories. Among the 6,034 and 6,069 AS genes in isolates Race 5 and Race 4, respectively significant overrepresentation (adjusted p-value< 0.01) of PFAM domains was observed for major facilitator superfamily (MFS) and sugar transporters, TFs of the fungal Zn(2)-cys(6) family, and cytochrome P450 enzymes (Fig 3A and 3B). Accordingly, among all the AS genes in both isolates, the most significantly enriched biological function GO terms were transmembrane transport (GO:0055085), regulation of transcription by RNA polymerase II (GO:0006357), carbohydrate transport (GO:0008643), and oxidoreductase activity (GO:0016491) (Fig 3C and S8 Table). Finally, based on hypergeometric tests, AS genes predicted to encode transporters (p-value< 1e-30), secreted proteins (p-value< 1e-8), cytochrome P450 enzymes (p-value< 1e-6), carbohydrate-active enzymes (CAZymes; p-value< 1e-6), and proteases (p-value< 0.01) were significantly overrepresented among all AS genes in both isolates (Table 2). In contrast, no significant enrichment or depletion of genes encoding candidate effectors was detected among the pool of AS genes in isolate Race 5 and/or Race 4. Collectively, these results indicate that AS in *C. fulvum* during host infections occurs more frequently in genes likely to be involved in the transport of sugars or other carbohydrates, regulation of genes, and monooxygenase activity, but less frequently in genes encoding proteins that are directly involved in modulation of host-immunity, such as effectors.

## AS genes are more abundantly present in repeat-rich chromosomes and exhibit longer upstream intergenic regions

We have previously determined that the genome of *C. fulvum* shows a compartmentalized architecture composed of gene-dense/repeat-poor regions interspersed with gene-sparse/

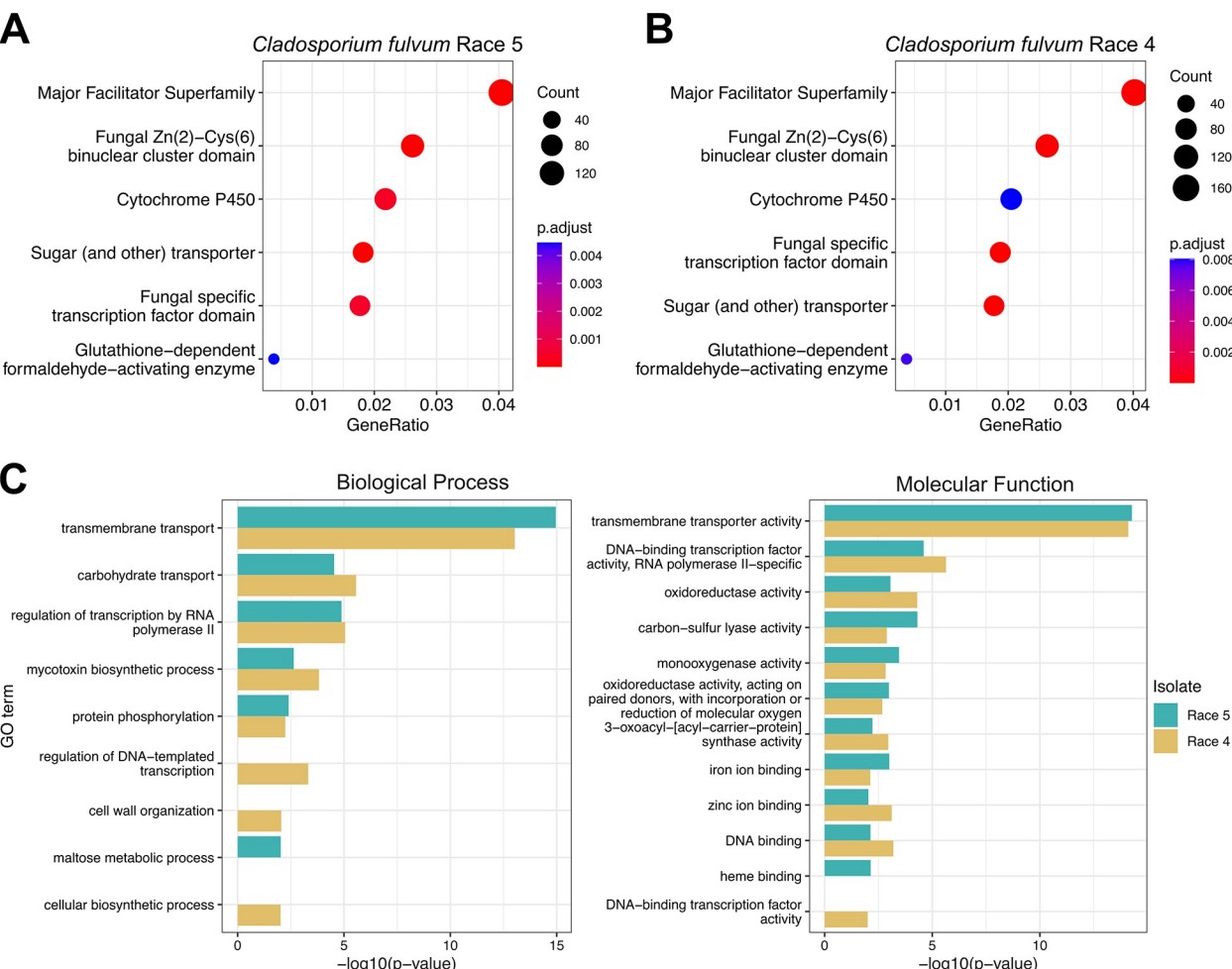

**Fig 3. Genes that are alternatively spliced (AS) in *Cladosporium fulvum* isolates Race 5 and Race 4 during tomato infections, are enriched for major facilitator superfamily (MFS) and sugar transporters, transcription factors, and cytochrome P450 enzymes.** (A-B) Dot plots showing the conserved PFAM domains that are significantly enriched among the AS genes in isolates Race 5 and Race 4. The size of the dots corresponds to the number of AS genes containing the respective PFAM domain. The x-axis shows what proportion of AS genes containing the respective PFAM domain contributes to the total of AS genes containing a conserved PFAM domain. Dots are color-coded based on enrichment p-values adjusted using the Benjamini–Hochberg method. (C) Bar charts showing p-values of gene ontology (GO) terms from the classes of biological process and molecular function that are enriched among the genes predicted to undergo AS in isolates Race 5 and Race 4. The x-axis indicates the enrichment p-values in negative log scale.

**Table 2. Genes that are alternatively spliced (AS) in *Cladosporium fulvum* isolates Race 5 and Race 4 during tomato infections are enriched for different functional categories.** The table shows the total number of genes and of AS genes in *C. fulvum* isolates Race 5 and Race 4 that are classified in the different gene functional categories. Enrichment p-values were obtained with hypergeometric tests.

| Functional category | Isolate Race 5 | | | Isolate Race 4 | | |
|---|---|---|---|---|---|---|
| | Total in the genome | Count of AS genes | p-value | Total in the genome | Count of AS genes | p-value |
| Transporters | 2293 | 1186 | 2.53E-33 | 2277 | 1189 | 4.80E-34 |
| Secreted proteins | 1416 | 677 | 1.20E-09 | 1407 | 673 | 4.63E-09 |
| Cytochrome P450 | 133 | 86 | 1.11E-08 | 134 | 83 | 2.03E-07 |
| CAZymes | 525 | 270 | 1.07E-07 | 519 | 268 | 2.75E-07 |
| Proteases | 362 | 167 | 0.013 | 358 | 171 | 0.004 |
| SM backbones | 42 | 22 | 0.076 | 41 | 23 | 0.041 |
| Candidate effectors | 432 | 182 | 0.233 | 431 | 180 | 0.298 |

repeat-rich regions [39,40]. We therefore examined whether the frequency and type of AS events was affected by the presence of the genes in one or the other region type. An inspection of the distribution of AS genes on the different chromosomes, showed that the number of AS genes varied among the core chromosomes, ranging between 33.9% and 33.6% of the total genes present in Chr4 of isolates Race 5 and Race 4, respectively to 45.7% and 45.8% of the total genes present in Chr1 of isolates Race 5 and Race 4, respectively (Figs 4A and S6 and S9 Table). Interestingly, in the dispensable Chr14, which is present in isolate Race 5 but absent in isolate Race 4, only 25.0% of the genes were AS. Overall, a positive correlation (Person correlation coefficient $r = 0.63$) was observed between abundance of AS genes and repetitive DNA content in the 13 core chromosomes (Fig 4B). For instance, in Chr3 which has the highest repetitive DNA content among core chromosomes (62%), 43.8% and 44.6% of the genes in isolates Race 5 and Race 4, respectively were AS during host infections. In contrast, in Chr13 which has the lowest repetitive DNA content (36%), only 36.9% of the genes in both isolates were AS. These results indicate that AS is more prevalent in genes located in repeat-rich regions of the genome and that dispensable chromosomes carry less AS genes.

Next, we investigated whether AS also affects more frequently genes present in gene-poor regions that are typically characterized by long intergenic regions and high amounts of repetitive DNA. To do so, we compared the distribution of the sizes of the intergenic regions for AS and non-AS genes. For both isolates Race 5 and Race 4, the upstream intergenic regions of AS genes (Race 5: mean = 3,911 bp, median = 759 bp; Race 4: mean = 3,873 bp, median = 762 bp) were significantly longer (Race 5: p-value< 2E-12; Race 4: p-value< 3E-13) compared to the upstream intergenic regions of non-AS (Race 5: mean = 3,290 bp, median = 684 bp; Race 4: mean = 3,313 bp, median = 682 bp) (Fig 4C). In contrast, no significant differences were observed when comparing the size of downstream intergenic regions of AS genes and non-AS genes in both isolates (Fig 4C). These results indicate that the upstream intergenic regions of AS genes, which include promoter and other *cis*-regulatory gene regions, were significantly longer compared to genes with no evidence of AS. Because long intergenic regions are in *C. fulvum* almost always associated with high repetitive DNA content [39], the amount of repetitive DNA was investigated. Indeed, the average repetitive DNA content of the upstream intergenic regions of AS genes was significantly (p-value< 0.01) larger in both isolates compared to non-AS genes (S7 Fig), but no significant difference in repetitive DNA content was observed for the downstream intergenic regions (S7 Fig). Finally, significant differences were observed between AS genes and non-AS genes with respect to their physical characteristics. Specifically, AS genes had an overall lower GC content, were longer, had more conserved PFAM domains, and had shorter exons compared to non-AS genes (S8 Fig and S10 Table). These observations suggest that gene structure influences AS frequency, potentially by facilitating secondary RNA structures and providing more opportunities for splicing events. The presence of more conserved domains in AS genes may also indicate their functional importance and evolutionary conservation.

## AS may putatively increase protein diversity in *C. fulvum* during tomato infections

AS has the potential to increase protein diversity in cells when it affects gene coding sequences, as opposed to AS events in 3' or 5' untranslated regions (UTRs). To investigate the extent to which AS theoretically increased protein diversity in *C. fulvum* during infections, ORFs in the assembled transcript isoforms of isolates Race 5 and Race 4 were predicted. A total of 26,632 and 26,108 ORFs could be predicted from the 26,818 and 26,397 unique transcript isoforms that were assembled from isolates Race 5 and Race 4, respectively. The sequences of the

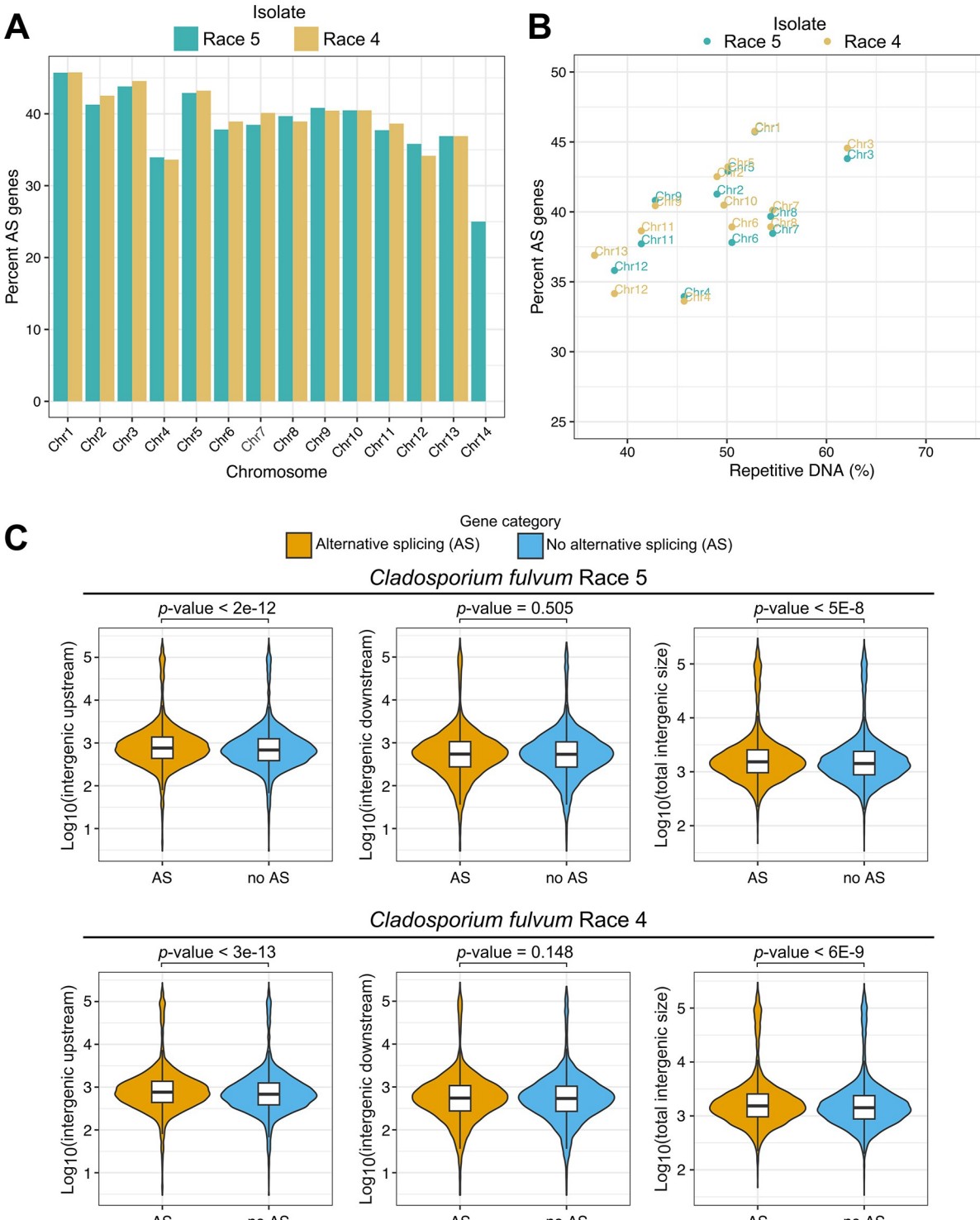

**Fig 4. Genes that are alternatively spliced (AS) in *Cladosporium fulvum* isolates Race 5 and Race 4 during tomato infections, are preferentially located in repeat-rich chromosomes and exhibit longer upstream intergenic regions compared to non-AS genes.** (A) Bar chart showing the percentage of AS genes (y-axis) present in each of the chromosomes (x-axis) of the reference genome of isolate Race 5. (B) Scatter plot showing that chromosomes with an overall higher repeat content typically contain higher percentages of AS genes, except for the dispensable chromosome Chr14 that is absent in isolate Race 4. Each point represents a chromosome, and points are color-coded to distinguish isolate Race 5 from isolate Race 4. (C) Violin plots showing the size distribution of the intergenic regions from AS and non-AS genes. The violin plots show that the upstream intergenic regions of AS genes are significantly longer compared to non-AS genes. P-values were obtained with the Wilcoxon rank sum test.

translated ORFs were subsequently organized into clusters with cd-hit such that identical or fully contained sequences were grouped together and each cluster represented a unique protein isoform. By doing so, a total of 19,757 and 19,551 protein isoforms were identified in isolates Race 5 and Race 4, respectively. These numbers are 31% higher than the predicted number of protein-encoding genes in the genomes of isolates Race 5 ($n$ = 14,993) and Race 4 ($n$ = 14,895) [40], suggesting that AS could theoretically increase protein diversity in *C. fulvum* during host infections.

From the 6,034 and 6,069 AS genes in isolates Race 5 and Race 4 respectively, 3,545 (58.7%) and 3,554 (58.5%) genes were predicted to produce distinct protein isoforms (S11–S13 Tables), indicating that the rest ~41% of the AS genes had splicing events in non-coding sequences. When considering this data in view of the entire genome, then only 23.6% to 23.9% of the total protein-coding genes in isolates Race 5 and Race 4, respectively contributed through AS to proteome diversity. A functional enrichment analysis indicated that genes encoding transporters (p-value< 4E-13), secreted proteins (p-value< 8E-9), and to a lesser extent CAZymes (p-value< 8E-4) and cytochrome P450 enzymes (p-value< 3E-3), were overrepresented in the pool of AS genes producing multiple protein isoforms in isolates Race 5 or Race 4 (S14 Table). For instance, 432 (12.2%) and 424 (11.9%) of such genes in isolates Race 5 and Race 4, respectively encoded secreted proteins (S11–S13 Tables). Interestingly, there were 134 effector-encoding genes among the AS genes yielding multiple protein isoforms, although on a genome-wide level no enrichment for genes encoding candidate effectors was observed (S14 Table). Included among the effector genes were the previously described *Ecp1* (S9 Fig) [42], *Ecp5* (S10 Fig) [43], *Ecp6* (S11 Fig) [44], and *Ecp12* effectors (S12 Fig) [45]. In particular, AS events modified the mature proteins of Ecp1, Ecp6, and Ecp12, while only the signal peptide was affected by an AS event in the gene encoding Ecp5. Analysis of isoform expression during interaction with tomato revealed that, in both Race 5 and Race 4 isolates, the longest isoform from *Ecp5* carrying 17 additional amino acids in the signal peptide, was the one that was preferentially expressed. Similarly, the shortest isoform from *Ecp6* that is missing six amino acids between the signal peptide and the first LysM conserved domain, was preferentially expressed. In contrast, none of the isoforms from *Ecp1* and *Ecp12* were clearly preferentially expressed during infection.

The vast majority of the AS genes producing multiple protein isoforms (i.e. 2,498 or 70.5% of the AS genes in isolate Race 5 and 2,540 or 71.5% of the AS genes in isolate Race 4) were predicted to yield just two isoforms (Fig 5A), and only 117 (3.3%) of the AS genes in isolate Race 5 and 134 (3.8%) of the AS genes in isolate Race 4 were predicted to encode five or more distinct protein isoforms (Fig 5A and S11–S13 Tables). Likewise, of the 2,950 AS genes shared by both isolates, 2,207 (74.8%) were predicted to yield a similar number of distinct protein isoforms, indicating a similar contribution to proteome diversity (Fig 5B and S11 Table). Two notable examples of such AS genes are *CLAFUR5_09979* and *CLAFUR5_09583*, which were predicted to yield 13 and 10 distinct protein isoforms in both isolates Race 5 and Race 4 (S13 Fig), and which encode putative TFs with homology to the CON7 TF required for appressorium formation and pathogenicity of *M. oryzae* [46,47], and to the ascospore maturation 1 protein (Asm-1) TF that regulates sexual and asexual reproduction in *Neurospora crassa* [48], respectively.

We finally examined the extent to which AS led to the gain or loss of conserved domains or of a signal peptide (SP) in the yielded protein isoforms (S11–S13 Tables). A total of 1,664 AS genes in isolate Race 5 and 1,841 AS genes in isolate Race 4 produced multiple protein isoforms that showed presence/absence variation in their PFAM domains (S15 Table). PFAM domains that varied the most among the protein isoforms were the major facilitator superfamily domain (PF07690; 67 AS genes in Race 5 and 81 AS genes in Race 4), the fungal Zn(2)-Cys (6) binuclear cluster domain (PF00172; 78 AS genes in Race 5 and 74 AS genes in Race 4), the

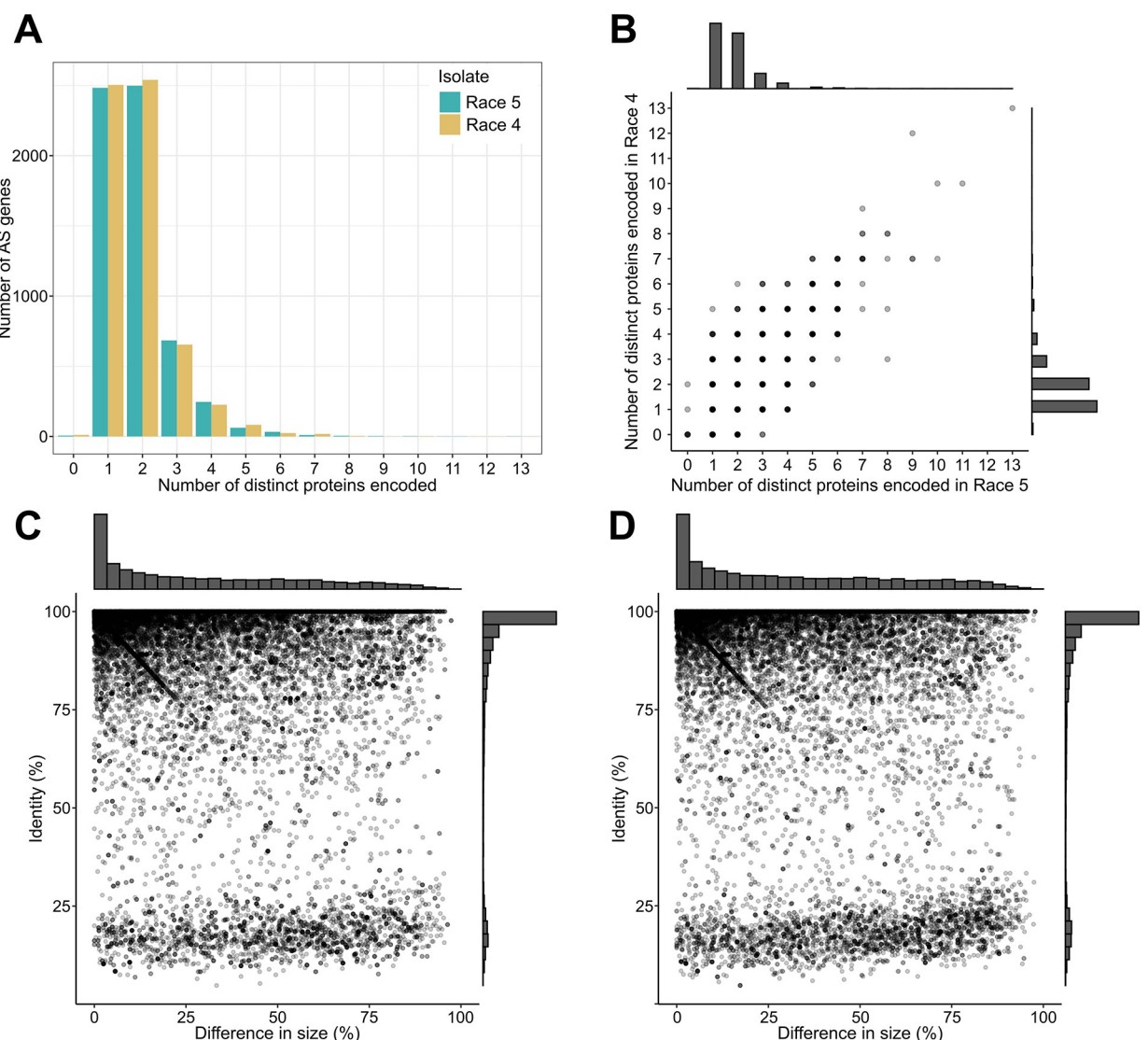

**Fig 5. Alternative splicing (AS) in pathogen genes during tomato infections putatively increases the protein diversity of *Cladosporium fulvum* isolates Race 5 and Race 4.** (A) Bar chart showing the number of AS genes in isolates Race 5 and Race 4 producing 0, 1, or more distinct protein isoforms (x-axis). (B) Scatter plot showing that a positive correlation exists among pairs of orthologous AS genes between isolates Race 5 and Race 4 that produce similar numbers of distinct protein isoforms. Each point represents a gene, and the plot shows a total of 6,493 AS genes in either isolate Race 5 or Race 4. (C-D) Scatter plots showing the diversity of protein isoforms produced by AS genes in isolate Race 5 (panel C) and Race 4 (panel D). Each point represents a pairwise alignment between the different protein isoforms that are produced by a single AS gene. Only AS genes predicted to yield two or more distinct protein isoforms are shown in the scatter plots. The y-axis shows the percent amino acid identity among the protein isoforms, and the x-axis shows the differences in size between the aligned protein isoforms as the percentage of the longest aligned isoform. Alignments were generated based on the local-global strategy, which is based on aligning the longest sequences locally and the shortest sequences globally.

short chain dehydrogenase domain (PF00106; 41 AS genes in Race 5 and 48 AS genes in Race 4), and the fungal specific TF domain (PF04082; 33 AS genes in Race 5 and 36 AS genes in Race 4). Along the same lines, 355 AS genes in Race 5 and 362 AS genes in Race 4 (intersection size = 270) yielded isoforms with presence/absence variation of SP (S15 Table). Included among these were 94 CAZyme and 76 effector encoding genes, including the previously described *Ecp30*, *Ecp42*, *Ecp53-1*, *Ecp33*, and *Ecp10-3* effectors [45].

## Differential isoform usage and isoform switching across different infection timepoints are mostly isolate-specific

A differential isoform usage (DIU) analysis was performed in order to detect statistically significant changes in the usage of the different transcript isoforms produced by the AS genes in isolates Race 5 and Race 4 during the course of tomato infection. Such changes could be functionally relevant, as they might signify the preferential production of isoforms with varying functional potential at different stages of the infection. For this purpose, the expression values of the 26,818 and 26,397 transcript isoforms assembled from isolates Race 5 and Race 4 were first estimated for each of the seven timepoints of the infection process that were sampled. Next, DIU analysis was carried out between all possible pairwise comparisons among the seven timepoints.

A total of 401 transcripts from 246 genes of isolate Race 5 and 166 transcripts from 103 genes of isolate Race 4, showed significant (p-value< 0.01) changes in their abundance across different timepoints of the infection (Fig 6A and 6B). Of the AS genes with significant DIU, 111 genes in isolate Race 5 and 42 genes in isolate Race 4 had AS events affecting their ORF, and thereby putatively yielded protein isoforms with altered levels of abundance during disease progression (S16 Table). Most of these genes (i.e. 76 and 34 genes in isolates Race 5 and Race 4, respectively) produced just two protein isoforms, but 35 genes in isolate Race 5 and seven genes in isolate Race 4 produced more than three isoforms. The majority also of the AS genes with DIU across the infection process encoded hypothetical proteins (i.e. 55 in isolate Race 5 and 26 in isolate Race 4), but 17 and 11 genes in isolate Race 5 and Race 4, respectively encoded for secreted proteins of which 2 and 2, respectively were candidate effectors (Fig 6C and 6D). Finaly, only 17 genes with DIU at the transcript level during disease progression were common to both isolates (S14 and S15 Figs and S16 Table) and only five had splicing events in their coding sequences and could thereby yield multiple protein isoforms. These five genes encoded a tRNA (uracil-O(2)-)-methyltransferase (*CLAFUR5_01082*), a lactose permease (*CLAFUR5_11255*), and three hypothetical proteins (*CLAFUR5_08245*, *CLAFUR5_09805*, *CLAFUR5_20329*), and resulted in the production of two protein isoforms each in both isolates Race 5 and Race 4 (*CLAFUR5_01082*, *CLAFUR5_08245*, *CLAFUR5_20329*), or in a different number of protein isoforms in the two isolates (*CLAFUR5_11255*, *CLAFUR5_09805*).

In addition to the DIU analysis, an isoform switching (IS) analysis was also carried out to identify pairs of transcript isoforms switching in their relative abundance during host infection. If such switches are present, then they would appear as an intersection in the transcript abundance of the different isoforms during disease progression. The analysis revealed a total of 193 and 123 pairs of transcript isoforms from 166 and 105 genes in isolates Race 5 and Race 4, respectively that significantly (p-value< 0.05) switched their expression values during host infection (S17 and S18 Tables). Of these, 13 (7.8%) genes in isolate Race 5 encoded secreted proteins and 2 (1.2%) encoded candidate effectors, while 17 (16.2%) genes in isolate Race 4 encoded secreted proteins and 7 (6.6%) candidate effectors. Overall, only 10 genes exhibited cases of IS in both isolates, which included the meiosis protein mei2 (*CLAFUR5_07241*), a polyubiquitin (*CLAFUR5_08768*), two hypothetical secreted proteins (*CLAFUR5_06894*, *CLAFUR5_14590*), an alcohol dehydrogenase (*CLAFUR5_01545*), an (R,R)-butanediol dehydrogenase (*CLAFUR5_06495*), a protoporphyrin uptake protein (*CLAFUR5_11018*), and three other hypothetical proteins (*CLAFUR5_08576*, *CLAFUR5_10847*, *CLAFUR5_12639*). Other genes with unusual isoform expression patterns include CLAFUR5_10677, which encodes a predicted 2-oxoglutarate-dependent ethylene/succinate-forming enzyme. The two AS isoforms of this gene exhibited contrasting expression trends between isolates Race 5 and Race 4, as in Race 5 their expression decreased over time, while in Race 4 it increased (S16 Fig). Another gene with contrasting isoform expression patterns is CLAFUR5_03540, which encodes a

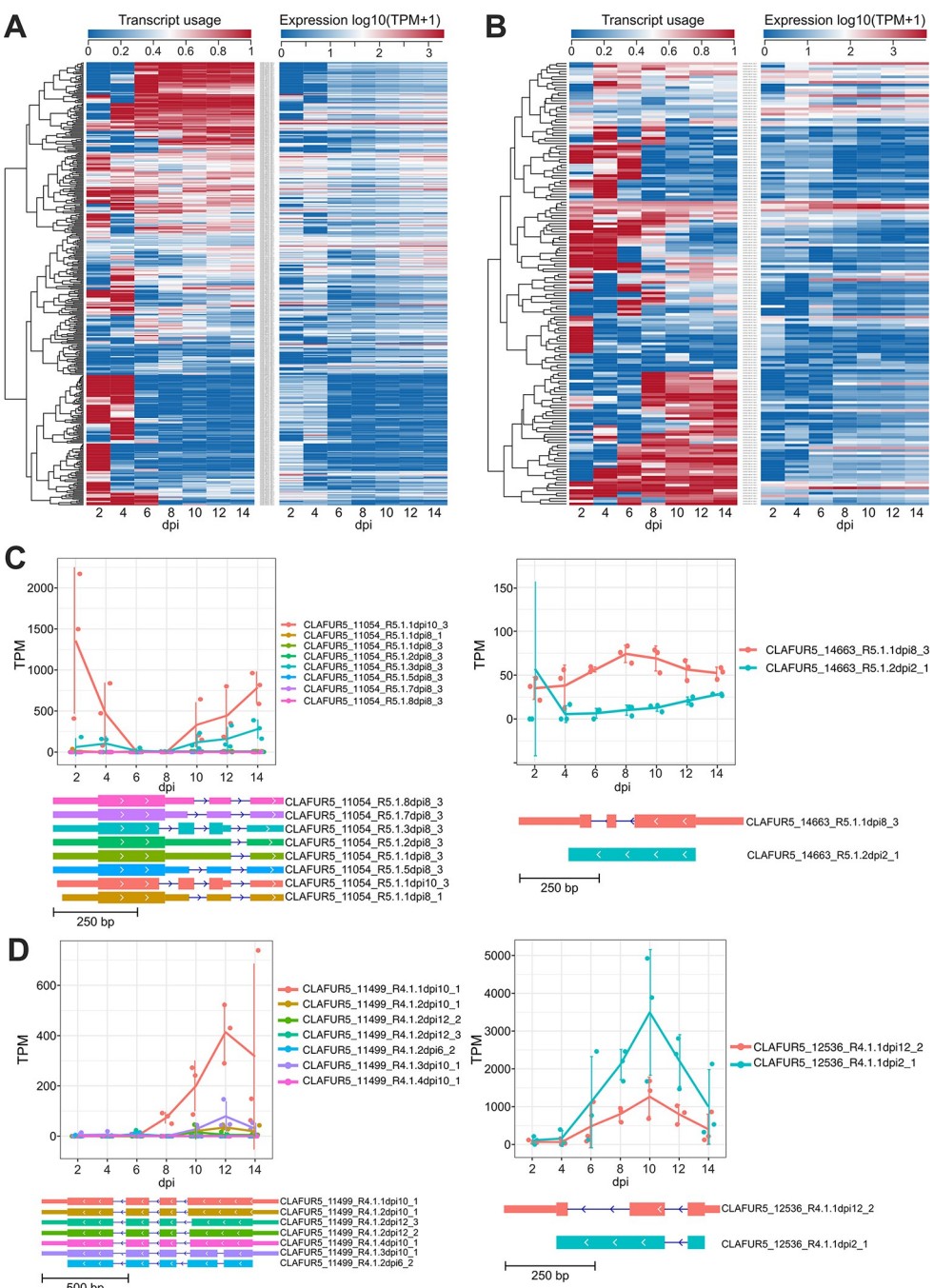

**Fig 6. Differential isoform usage (DIU) in alternatively spliced (AS) genes of *Cladosporium fulvum* isolates Race 5 and Race 4 during the time course of tomato infections.** (A-B) Transcript usage and relative expression of the 401 transcripts from 246 genes of isolate Race 5 (panel A) and 166 transcripts from 103 genes of isolate Race 4 (panel B) with changes in abundance, when originating from the same gene, during different timepoints of the infection. In both panels A and B, the left hand-side heat map shows transcript usage as the fraction of the sum of the expression of all transcripts from the gene and considering the average expression values of three replicates. The heat map on the right hand-side shows the expression in transcripts per million (TPM) for the transcripts. (C-D) Examples of AS genes with DIU during the infection, encoding candidate effectors from *C. fulvum* isolates Race 5 and Race 4. The line graphs show two AS genes (i.e. CLAFUR5_11054, CLAFUR5_14663) from isolate Race 5 (panel C) and two AS genes (i.e. CLAFUR5_11499, CLAFUR5_12536) from isolate Race 4 (panel D). In the line graphs, the points represent the expression values in TPM (transcripts per million) of the individual transcripts at different timepoints of the infection. Standard deviation in the TPM values from three infections (i.e. biological replicates) is shown as vertical lines. The trends of transcript expression across time are shown as thick lines connecting the average TPM values for each

individual transcript. The exon/intron structures of the transcripts are shown at the bottom of each line graph, with the predicted coding sequences represented as thicker boxes.

putative fructose-bisphosphate aldolase. This gene produces three AS isoforms and while expression of one of its isoforms showed a S-shaped pattern in isolate Race 5, it exhibited a V-shaped pattern in isolate Race 4 (S16 Fig).

It is currently unknown whether IS in these genes or genes with DIU has any functional consequences for infections or if they represent transcriptional noise. However, given the low number of genes with DIU or IS common to the two isolates, it can be assumed that their potential impact of on host infections may be isolate-specific rather than at the species level.

## Discussion

In this study, we systematically analyzed the landscape of AS events occurring in genes of two *C. fulvum* isolates, i.e. Race 5 and Race 4, during a complete fungal infection cycle on the tomato host. Our transcriptomic analyses revealed a significant degree of heterogeneity in the transcript isoforms assembled from different isolates, infections, and infection timepoints, suggesting that the majority are the result of stochastic noise in the transcriptional and splicing machinery. However, given the dynamic nature of AS in cells and the complexity of the splicing machinery, stochastic fluctuations in the splicing output are to an extent expected [49]. Other factors that are difficult to control, such as inaccuracies in sequencing and transcriptome assembly, cell-to-cell variability in splicing and transcription kinetics, and others [50–52], could contribute to the transcriptional noise and the discrepancies observed in the splicing outputs as well. Despite the high sample-to-sample heterogeneity in the assembled transcript isoforms, our studies showed that ~40% of the protein-coding genes in each of the two isolates of *C. fulvum* were AS in more than one sample, with ~37.5% of them being AS in both isolates Race 5 and Race 4. This indicates that a sizeable fraction of C. *fulvum* genes are recurrently AS across different infections and/or isolates, suggesting that splicing in these cases may be functionally relevant. The percentage of AS genes in *C. fulvum* is also very high compared to reports in other fungal pathogens, which typically have less than 30% of their genes undergoing AS [18,19,21,33,49,53]. Yet, most studies, including this one, are in agreement that IR is the most frequent type of AS in fungi [21,29,33,35,49,54] and in contrast to mammals, in which ES is typically the prevalent type of AS [55].

It was previously shown that the genome of *C. fulvum* is rich in TEs and exhibits a bipartite architecture that resembles the 'two-speed genome' model of evolution, with candidate effector genes enriched in TE-rich regions [39]. Here, we found that AS genes were more frequently present in repeat-rich chromosomes and have significantly longer upstream intergenic regions with higher repetitive DNA content. An exception the dispensable Chr14, as the majority of the genes in this chromosomes are transcriptionally inactive during interaction with tomato [40]. The insertion of TEs in intergenic or intronic regions has been previously associated with changes in AS patterns in plants and humans [56–59], and it is thus plausible that a connection exists between TEs and the induction of AS in genes of *C. fulvum* as well. It would be interesting to examine whether this applies more generally to other fungal species and the extent to which AS associates with the organization of their genomic content.

Recently, a comparative analysis of AS in seven human fungal pathogens showed that genes subjected to AS during host infections were mostly associated with the functionality of the cell membrane, whereas AS under environmental stress conditions mainly affected genes with diverse regulatory functions [32]. Likewise, in the rice-blast fungus *M. oryzae*, it was shown that genes that were AS during infections were mostly enriched for TFs and phospho-transfer-ases (35). These and various other studies have highlighted that different gene categories are

differentially affected by AS in response to similar stresses, and that the complement of genes undergoing AS in fungi varies significantly among species [19,20,32]. In our studies, we found that in *C. fulvum*, AS during host infections frequently affects genes encoding TFs, suggesting that it may have a predominantly regulatory effect by reprogramming gene expression. The genes *CLAFUR5_09583* and *CLAFUR5_09979* were notable examples, as they encode orthologs of the ASM-1 [48] and CON7 [46] TFs, respectively and each is predicted to yield ten or more distinct protein isoforms. Recently, CON7 was shown to be a key transcriptional regulator in *Fusarium graminearum*, affecting genes involved in conidiation, sexual development, virulence, and vegetative growth [60], whereas ASM-1 is shown to affect morphogenesis (e.g. conidiation) and development in several fungal species [61]. The contribution of *CLAFUR5_09583* and *CLAFUR5_09979* in virulence of *C. fulvum* is unknown, but since both genes were AS in both isolates during infection and were predicted to yield the same number of protein isoforms, it is possible that some of the produced isoforms are biologically meaningful for infections. Overall, the prevalence of splicing in TF-encoding genes supports that AS could have a role in *C. fulvum* in modulating infections on its tomato host.

Other functional gene categories enriched among AS genes in *C. fulvum* were MFS transporters, sugar transporters, and cytochrome P450 enzymes. The functional significance of AS in these gene categories is perhaps less clear but given their general involvement in nutrition, metabolism, and cellular detoxification processes, it may imply that AS in these genes promotes adaptation to the host environment. MFS transporters form the largest superfamily of secondary active transporters that collectively transport a broad spectrum of substrates, thereby participating in diverse physiological processes and stress responses, including nutrient acquisition, resistance against oxidative stress and xenobiotic compounds, secretion of endogenously produced toxins, and others [62–65]. Likewise, fungal sugar transporters are involved in the uptake of small plant-derived sugar molecules [66] and they may further have functions in sugar sensing, carbon catabolite repression, and utilization of the most favorable carbon source in the environment [67–70]. Finally, cytochrome P450 monooxygenases are a diverse superfamily of proteins known to be involved in cellular metabolism, xenobiotic detoxification, synthesis of toxins [71,72], and other metabolic processes of relevance to infections [73–75]. Collectively, the above suggest that in *C. fulvum*, AS in MFS transporters, sugar-like transporters, and cytochrome P450 enzymes during tomato infections may offer a means to augment and fine-tune virulence on the host at multiple levels, including metabolic adaptation to the host's carbon and nutrient environment, protection against oxidative stress and plant defense compounds, production of toxins during pathogenesis, and others. However, despite the possible contribution of these gene families to infections, in contrast, genes encoding proteins that are directly involved in modulation of host-immunity, such as effectors, were less frequently affected by AS. Though, caution should be drawn here as many effectors are typically expressed at very early stages of the infection process when fungal biomass is still very limited and the detection of fungal transcripts in the sequenced samples challenging. Therefore, it is perhaps not surprising that genes involved in various metabolic processes were more readily detected in the pool of AS genes, as compared to genes encoding effectors.

By comparing the expression of pathogen-derived transcripts at seven timepoints during host infections, we identified several cases of DIU in AS genes of *C. fulvum*. Most of these genes encoded hypothetical proteins, but a few encoded for effector suggesting that infection stage-dependent modulation of AS in effector-encoding genes could possibly prime infections of the host by the selective production or downregulation of specific functional isoforms of the effectors. Finally, ~41% of the AS genes in *C. fulvum* had splicing events in their 5' or 3' UTRs, thereby producing just a single protein isoform. Such splicing events, although they do not increase protein diversity, increase the functional diversity in the 5' or 3' UTRs that could lead

to significant alterations in gene expression, protein translation and localization [76–80]. Collectively, the high frequency of AS events in the 5' or 3' UTRs of *C. fulvum* genes during infections may suggest an additional level of post-transcriptional control of the infection process that so far remains largely unexplored.

## Materials and methods

### Inoculations of tomato plants with *C. fulvum* isolates

*Cladosporium fulvum* isolates Race 5 [81] and Race 4 [82] were grown in half-strength potato dextrose agar (PDA) for two weeks at 23˚C. Tomato plant (*Solanum lycopersicum* cv. Moneymaker) inoculations were performed as previously described [83,84]. Briefly, *C. fulvum* spores were collected from two-week-old PDA plates growing at 25˚C in the dark. Moneymaker tomato plants were grown for six weeks in a growth chamber with 16 hr light, 70% humidity at 25˚C, and 8 hr dark, 90% humidity at 23˚C. $10^6$/ml spores were suspended in 10% potato dextrose broth (PDB) and sprayed on the lower leaf side of ten six-week-old tomato plants. The inoculated plants stayed in the dark, with 98% humidity for the first two days. After that, they were returned to the 16/8 hrs light/dark conditions as indicated above. Two infected leaves from each inoculated plant were harvested at 2, 4, 6, 8, 10, 12, and 14 dpi. Collected samples were immediately frozen in liquid nitrogen and stored at -80˚C until RNA extraction. The inoculation was repeated three times under the same conditions.

### RNA extraction and sequencing

Samples were ground to a powder in liquid nitrogen, and total RNA was extracted using Trizol (Invitrogen, Carlsbad, CA, USA). RNA quality was measured using a Qubit fluorometer (Life Technologies, New York, NY, USA) and the Bioanalyzer 2100 (Agilent Technologies, Santa Clara, CA, USA). Preparation and sequencing of the polyA-selected RNAseq libraries were outsourced to the DNA Technologies and Expression Analysis Core Laboratory at the UC Davis Genome Center (https://dnatech.genomecenter.ucdavis.edu/). Libraries were sequenced on an Illumina NovaSeq 6000 instrument (PE150 format).

### Comparison of introns and splicing sites

The number of introns supported by RNAseq reads in *C. fulvum* Race 4 was obtained by analyzing the splicing junction table generated by STAR v2.7.9a, after mapping the RNAseq reads to the genome of *C. fulvum* Race 4 [40]. To estimate conservation of number and size of introns, one-to-one pairs of orthologous genes from *C. fulvum* Race 5 and Race 4 were obtained with OrthoFinder v.2.5.4 [85]. Number and size of introns of orthologous genes were then investigated with support of the script *agat_sp_add_introns.pl* from AGAT v1.2 package [86] to add introns to the gene annotation files. To investigate whether the ortholog genes share the same intron start and end coordinates, the gene annotation of isolate Race 4 was mapped to the genome of isolate Race 5 using Liftoff v1.6.3 [87]. This resulted in a new annotation with coordinates of genes, exons, and introns from isolate Race 4 in the genome of isolate Race 5. The new annotation generated by this procedure allowed direct comparison of the intron coordinates based on the reference genome of isolate Race 5.

### Estimation of percentages of RNAseq reads from *C. fulvum* and tomato genomes

Prior to read mapping, the RNAseq reads were processed with the script *bbduk.sh* from BBMap package v38.90 [88] to remove remaining adapter sequences (parameters: *ktrim = r*,

*k = 23*, *mink = 11*, *hdist = 1*, *minlength = 40*, *tpe*, and *tbo*). To quantify and remove reads that originated from tomato, the trimmed reads were processed with *seal.sh* from BBMap package v38.90 (parameters: *ambiguous = toss*, and *k = 27*), using as reference the genomes of *C. fulvum* Race 5 [39] (GenBank accession GCA_020509005.2) and *Solanum lycopersicum* version SL4.0 [89]. Briefly, RNAseq reads that had better match to the *S. lycopersicum* genome instead of the *C. fulvum* genome and reads that were equally well matched to both genomes, were filtered out. RNAseq reads that had a better match to the *C. fulvum* genome and reads considered unmatched to any genome were used for downstream analysis.

## Read mapping and transcriptome assembly

The filtered RNAseq reads were mapped to the *C. fulvum* Race 5 genome with STAR v2.7.9a [90] in 2-pass mode (parameters:—*twopassMode Basic*,—*alignIntronMin 20*,—*alignIntronMax 2000*,—*outSAMtype BAM SortedByCoordinate*). The mapped reads were assembled into full length transcripts with StringTie v2.1.7 [91], using the gene annotation of *C. fulvum* Race 5 as reference (parameter: -*G*). Because the genome of *C. fulvum* Race 5 has many genes physically close to each other, with median intergenic region size of only 646 bp [39], overlapping of untranslated regions (UTRs) between neighboring genes is common, which can result in chimeric transcripts during assembly. To minimize this issue, a gene-by-gene transcriptome assembly strategy was utilized. This strategy consisted of assembling the transcripts for each individual sample by extracting the RNAseq reads from the respective sample mapped to the annotated gene space using SAMtools v1.9 [92], and then assembled into full length transcripts using StringTie, which is considered as one of the best reference-based transcriptome assemblers, despite showing overall low precision levels of between 29% and 59% at the transcript level [93,94]. By doing so, transcripts for each gene were obtained for each sample (7 time-points x 3 replicates x 2 isolate = 42). To facilitate downstream analyses, assembled transcripts were assigned IDs that contained the name of the sample from which they originated.

## Detection of alternative splicing genes

Using the GTF files of the assembled transcripts and the reference genomes of *C. fulvum* Race 5, transcript sequences were extracted with gffread v0.12.7 [95]. The sequences were then clustered with cd-hit v4.8.1 (parameters: -*T 8 -M 2048 -c 1 -d 0*) [96], such that identical or fully contained transcripts were organized into the same cluster. Thus, each cluster represents a unique transcript. A table with the clusters were obtained with the script *cluster2txt* that comes with cd-hit. The organization of transcripts into clusters allowed to identify whether the cluster included transcripts assembled using RNAseq reads from a specific biological replicate from a timepoint. Specifically, the representative transcript *t* of the cluster *c* was considered present in a sample *s* if there was at least one transcript *u* assembled using reads from *s* such that *u* was present in *c*. This strategy allowed the identification of transcripts supported by multiple replicates, and whether they were shared or unique to the isolates Race 5 and Race 4. Because transcripts supported by only one sample are likely random transcriptional events, only transcripts supported by at least two samples, i.e., transcripts that could be replicated, were considered for downstream analyses. After that, one additional filtering step was applied. First, the script *agat_sp_add_introns.pl* from AGAT v1.2 package [86] was used to add intron coordinates to the GTF files containing the assembled transcripts. Then, the intron coordinates were compared among isoforms using a custom bash script. The isoforms containing the same number of introns and the same coordinates were considered duplicates, and only the longest one was kept. Finally, genes were considered undergoing AS, if they encoded at least two distinct

transcripts that remained after the filtering steps. The steps to assemble the transcripts and identify AS genes are summarized in S17 Fig.

## Classification of AS types and gene enrichment

AS events were classified into Skipping Exon (SE), alternative 5'/3' Splice Sites (A5/A3), mutually exclusive exons (MX), intron retained (IR), and alternative first/last exons (AF/AL) with the command *generateEvents* from SUPPA v2.3 [97] with default settings, except for IR events, which were identified with the "variable boundary" parameter (*-b V*) set to 50 to relax the restrictive default behavior of SUPPA2 to identify IR events. Gene enrichments were performed for conserved PFAM domains, GO terms, and functional gene categories. Enrichment for PFAM domains was conducted with clusterProfiler v.4.6.2 [98] with Benjamini-Hochberg adjusted p-value threshold of 0.05 based on PFAM domains identified with InterProScan v5.59–91.0 [99]. Enrichment for GO terms was performed with topGO v2.52 [100] with p-value threshold of 0.01 based on GO terms identified with PANNZER2 [101], using minimum Positive Predictive Value (PPV) of 0.4. Enrichment of functional gene categories was performed with hypergeometric tests using the R function *phyper* based on the lists of gene category reported previously [40]. All three types of enrichment were conducted within R v4.3.1.

## Prediction of ORFs and functional impact of AS

Before predicting ORFs within the transcripts, the transcripts in GFF format obtained from Race 4 using Race 5 genome as reference, were mapped to the genome of Race 4 using liftoff v1.6.3 [87] with default settings. By doing so, a new GFF file with transcripts coordinates in the genome of Race 4 was obtained. The nucleotide sequences of the transcripts from Race 5 and Race 4 were extracted using gffread v0.12.7 [95]. ORFs were predicted with ORFanage v1.2.0 [102] using parameters adjusted to finding ORFs of at least 120 bp that best matched the gene annotation (parameters:—*best* and—*minlen 120*). Transcripts with no predicted ORF were further processed with ORFfinder v0.4.3 with parameters adjusted to predict ORFs of at least 120 bp and starting only with ATG (parameters: *-s 0*, *-ml 120*, *-strand plus*). Only the longest ORFs predicted by ORFfinder were retained. For each isolate Race 5 and Race 4, the predicted protein sequences were clustered with cd-hit v4.8.1 [96] such that identical or fully contained protein sequences were present in the same group (parameter *-c 1*). Genes encoding distinct proteins were identified based on the cd-hit results. Specifically, the protein sequences encoded by the isoforms of a gene grouped in distinct cd-hit clusters, then the gene was considered to encode distinct proteins. To investigate to what extent protein sequences encoded by isoforms of the same gene differ in amino acid sequence, the protein sequences encoded by isoforms were aligned in a pairwise manner using the *pairwiseAlignment* function within the R package Biostrings v2.68.1 [103] using the "local-global" alignment strategy, such that the gaps at the end of the alignment do not penalize the alignment score. To investigate gain or loss of conserved motifs among isoforms, conserved PFAM domains within the protein sequences were identified with InterProScan v5.59–91.0 [99]. Presence/absence variation of PFAM domains among protein isoform was obtained by analyzing the output of InterProScan using a custom R script within R v4.3.1. Proteins with signal peptide were predicted using SignalP v6 [104]. AS genes encoding at least one protein with a predicted signal peptide and at least one protein without a signal peptide were considered to exhibit gain or loss of signal peptide. Functional annotations of protein isoforms were obtained by querying the protein sequences against the SwissProt database [105] with BLASTp (parameters: *-evalue 1e-10 -outfmt 6 -num_alignments 1 -max_hsps 1*). Protein isoforms encoding CAZymes were identified with dbCAN3 [106]. Candidate effectors were identified as described in Zaccaron and Stergiopoulos, 2024 [40].

The 3D structures of encoded mature proteins of the isoforms of the effectors *Ecp1*, *Ecp5*, *Ecp6*, and *Ecp12* were predicted with ColabFold v1.5.5 [107] using AlphaFold2 and MMseqs2.

## Transcript quantification and differential isoform usage (DIU)

To estimate expression levels of the assembled transcripts, first their nucleotide sequences were extracted with BEDtools v2.29.0 [108] and used to build an index with Salmon v1.10.0 [109] (parameter—*keepDuplicates*). After that, Salmon v1.10.0 was used in mapping-based mode (parameters: *-l IU*,—*gcBias*,—*seqBias*) to calculate the expression levels of the transcripts using the reads after trimming with *bbduk.sh* and selecting those with a *k*-mer matching with the *C. fulvum* genome. By doing so, Salmon generated expression values in Transcripts Per Million (TPM), which were used in SUPPA2 v2.3 [97] to predict differential transcript usage among all possible pairs of timepoints following SUPPA2's specification. More precisely, iso-form inclusion levels were quantified with the command *psiPerIsoform* using the TPM values, followed by the command *diffSplice* to calculate differential splicing between conditions with replicates (parameters:—*area 1000*,—*lower-bound 0.05*,—*combination*,—*tpm-threshold 2*,—*gene-correction*). A filtering step was carried out to keep only events with p-value < 0.01 and differential splicing value (dPSI) > 0.2. Isoform switch events were detected with the Time-Series Isoform Switch (TSIS) program [110] using the *mean* method to identify intersection points, p-value< 0.05, and isoform switch probability cutoff< 0.5.

## Supporting information

**S1 Text. RNA sequencing of *C. fulvum* isolates Race 5 and Race 4 during interaction with tomato.**
(PDF)

**S2 Text. Isolates Race 5 and Race 4 of *C. fulvum* share nearly all their intron splice sites.**
(PDF)

**S3 Text. Transcriptome profiling of *C. fulvum* during host infections reveals extensive transcript isoform heterogeneity among isolates and infections.**
(PDF)

**S1 Fig. Preliminary transcriptome assembly generated chimeric transcripts spanning genes physically close in the genome.**
(PDF)

**S2 Fig. A high heterogeneity in transcripts produced by *Cladosporium fulvum* isolates Race 5 and Race 4 during tomato infections is seen between the two isolates and the three different infections that were performed with each isolate.**
(PDF)

**S3 Fig. An overall low number of transcripts are constitutively present in samples from all three tomato infections (i.e. biological replicates) performed either with *Cladosporium fulvum* isolate Race 5 or isolate Race 4, and in every of the seven infection timepoints that were sampled per infection.**
(PDF)

**S4 Fig. The number of transcripts shared by *Cladosporium fulvum* isolates Race 5 and Race 4 increases when singleton transcripts were filtered out.**
(PDF)

**S5 Fig. The number of transcripts present in samples from all three tomato infections performed either with *Cladosporium fulvum* isolate Race 5 or Race 4, and in each of the seven infection timepoints that were sampled per infection, after filtering out singleton transcripts that were present in only one sample.**
(PDF)

**S6 Fig. The genomic distribution of genes predicted to be recurrently alternative spliced (AS) in *Cladosporium fulvum* isolates Race 5 and Race 4 during tomato infections.**
(PDF)

**S7 Fig. The upstream genomic regions of alternative spliced (AS) genes in *Cladosporium fulvum* isolates Race 5 and Race 4, have higher amounts of repetitive DNA compared to the upstream genomic regions of non-AS genes.**
(PDF)

**S8 Fig. Genes that are alternatively spliced (AS) in *Cladosporium fulvum* isolates Race 5 and Race 4 during tomato infections exhibit distinct physical characteristics as compared to non-AS genes.**
(PDF)

**S9 Fig. Alternative splicing (AS) in the effector gene *Ecp1* of *Cladosporium fulvum* isolates Race 5 and Race 4.**
(PDF)

**S10 Fig. Alternative splicing (AS) in the effector gene *Ecp5* of *Cladosporium fulvum* isolates Race 5 and Race 4.**
(PDF)

**S11 Fig. Alternative splicing (AS) in the effector gene *Ecp6* of *Cladosporium fulvum* isolates Race 5 and Race 4.**
(PDF)

**S12 Fig. Alternative splicing (AS) in the effector gene *Ecp12* of *Cladosporium fulvum* isolates Race 5 and Race 4.**
(PDF)

**S13 Fig. Two genes encoding putative transcription factors in *Cladosporium fulvum* isolates Race 5 and Race 4, and their predicted protein isoforms produced via alternative splicing (AS) events.**
(PDF)

**S14 Fig. Genes from *Cladosporium fulvum* isolate Race 5 with significant evidence of differential isoform usage at the transcript level during disease progression, which are common to both isolates.**
(PDF)

**S15 Fig. Genes from *Cladosporium fulvum* isolate Race 4 with significant evidence of differential isoform usage at the transcript level during disease progression, which are common to both isolates.**
(PDF)

**S16 Fig. Examples of genes from *Cladosporium fulvum* producing isoforms with unsual expression patterns during the infection process.**
(PDF)

**S17 Fig. Flowchart summarizing the steps performed to assemble transcripts of *Cladosporium fulvum* isolates Race 5 and Race 4, and identify genes predicted to undergo alternative splicing.**
(PDF)

**S1 Table. Number (No.) of raw paired-end reads obtained for *Cladosporium fulvum* isolates Race 5 and Race 4, during interaction with tomato (*Solanum lycopersicum*) cv. Moneymaker.**
(XLSX)

**S2 Table. Number and percentage of paired-end reads obtained for *Cladosporium fulvum* isolates Race 5 and Race 4 that had a k-mer matching to the genomes of *C. fulvum* or tomato (*Solanum lycopersicum*) cv. Moneymaker.**
(XLSX)

**S3 Table. The number (No.) of protein-coding genes in the genomes of *Cladosporium fulvum* isolates Race 5 and Race 4 with the specified number of introns.**
(XLSX)

**S4 Table. Number (No.) of transcript isoforms assembled for *Cladosporium fulvum* isolates Race 5 and Race 4, at each of the seven timepoints during interaction with tomato.**
(XLSX)

**S5 Table. Number of unique transcript isoforms assembled from alternatively spliced (AS) genes in *Cladosporium fulvum* isolates Race 5 or Race 4.**
(XLSX)

**S6 Table. The subset of alternatively spliced (AS) genes that are common between *Cladosporium fulvum* isolates Race 5 and Race 4.**
(XLSX)

**S7 Table. Number (No.) and percentages of alternative splicing (AS) events in *Cladosporium fulvum* isolates Race 5 and Race 4.**
(XLSX)

**S8 Table. Enrichment analysis of alternatively spliced (AS) genes in *Cladosporium fulvum* isolates Race 5 and Race 4.**
(XLSX)

**S9 Table. Number and percentages of genes from *Cladosporium fulvum* isolates Race 5 and Race 4 harboring different types of alternative splicing (AS) events.**
(XLSX)

**S10 Table. Mean and median values of the distributions of gene size, GC content, number of PFAM domains, and size of exons of alternatively spliced (AS) genes as compared to non-AS genes in *Cladosporium fulvum* isolates Race 5 and Race 4 during tomato infections.**
(XLSX)

**S11 Table. Alternatively spliced (AS) genes in *Cladosporium fulvum* isolates Race 5 and Race 4, producing multiple distinct protein isoforms.**
(XLSX)

**S12 Table. Functional annotation of the protein isoforms that are produced by alternatively spliced (AS) genes of *Cladosporium fulvum* isolate Race 5.**
(XLSX)

**S13 Table. Functional annotation of the protein isoforms that are produced by alternatively spliced (AS) genes of *Cladosporium fulvum* isolate Race 5.**
(XLSX)

**S14 Table. Enrichment analysis of alternatively spliced (AS) genes in *Cladosporium fulvum* isolates Race 5 and Race 4 producing multiple distinct protein isoforms.**
(XLSX)

**S15 Table. Alternatively spliced (AS) genes in *Cladosporium fulvum* isolates Race 5 and Race 4, producing multiple distinct protein isoforms with presence/absence variation in PFAM domains and signal peptides (SP).**
(XLSX)

**S16 Table. Alternatively spliced (AS) genes in *Cladosporium fulvum* isolates Race 5 and Race 4, producing transcript isoforms with (Yes) or without (No) evidence of differential isoform usage across the seven sampled timepoints of the infection process.**
(XLSX)

**S17 Table. Isoform switch events in alternatively spliced (AS) genes in *Cladosporium fulvum* isolate Race 5 during infection process.**
(XLSX)

**S18 Table. Isoform switch events in alternatively spliced (AS) genes in *Cladosporium fulvum* isolate Race 4 during infection process.**
(XLSX)

## Acknowledgments

We are thankful to Jonathan Niño-Sánchez and Anthony Salvucci for aiding with the plant inoculations.

## Author Contributions

**Conceptualization:** Alex Z. Zaccaron, Ioannis Stergiopoulos.

**Data curation:** Alex Z. Zaccaron.

**Formal analysis:** Alex Z. Zaccaron.

**Funding acquisition:** Ioannis Stergiopoulos.

**Investigation:** Alex Z. Zaccaron, Li-Hung Chen.

**Methodology:** Li-Hung Chen.

**Resources:** Ioannis Stergiopoulos.

**Software:** Alex Z. Zaccaron.

**Supervision:** Ioannis Stergiopoulos.

**Validation:** Alex Z. Zaccaron.

**Visualization:** Alex Z. Zaccaron.

**Writing – original draft:** Alex Z. Zaccaron, Ioannis Stergiopoulos.

**Writing – review & editing:** Alex Z. Zaccaron, Li-Hung Chen, Ioannis Stergiopoulos.

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
