## [Decision Letter · Decision Letter 0]

17 Sep 2024

Dear Prof. Dr Stergiopoulos,

Thank you very much for submitting your manuscript "Transcriptome analysis of two isolates of the tomato pathogen Cladosporium fulvum, uncovers genome-wide patterns of alternative splicing during a host infection cycle." for consideration at PLOS Pathogens. As with all papers reviewed by the journal, your manuscript was reviewed by members of the editorial board and by several independent reviewers. In light of the reviews (below this email), we would like to invite the resubmission of a significantly-revised version that takes into account the reviewers' comments.

The work was evaluated by three experts in the field, who generally found it interesting and thus of potential value to the community. However, several concerns were raised regarding the analyses that were performed, and on this basis the response from the reviewers was mixed, with one reviewer recommending rejection, and the other two recommending either minor or major revision.  Therefore, some additional experimentation and/ or computational analysis is likely required to shore up the conclusions of the paper and aid the reviewers in reaching a consensus recommendation.

We cannot make any decision about publication until we have seen the revised manuscript and your response to the reviewers' comments. Your revised manuscript is also likely to be sent to reviewers for further evaluation.

Sincerely,

Richard A Wilson

Academic Editor

PLOS Pathogens

Bart Thomma

Section Editor

PLOS Pathogens

Michael Malim

Editor-in-Chief

PLOS Pathogens

orcid.org/0000-0002-7699-2064

Reviewer's Responses to Questions

**Part I - Summary**

Reviewer #1: alternative splicing is an aspect of fungal pathogenomics that has not been well-studied so far... and this study is an interesting contribution to the knowledge of this topic

the manuscript was well written and of good quality overall

Reviewer #2: The manuscript by Zaccaron et al examines alternative splicing during a host infection cycle for two different isolates of the tomato pathogen Cladosporium fulvum. The authors have generated a highly interesting dataset that will be of interest not just for the study of alternative splicing but also for examining transcriptome dynamics in the species. The analyses are all expertly done and the manuscript very clearly written.

Reviewer #3: The study examines alternative splicing (AS) in the two fungal isolates, Race 4 and 5, of Cladosporium fulvum at seven timepoints of infection in tomato. The results revealed a relatively high percentage of AS genes in this fungus and overall similar patterns of AS in the two isolates. Interestingly, the study found that AS genes were enriched in repeat-rich chromosomes and had longer upstream intergenic regions. The exception was found in a dispensable chromosome. Differential isoform usage (DIU) for all pair-wise comparisons were performed and isoform switching events were identified.

AS analysis is not a straightforward process. The approaches used well reasonably selected.

**Part II – Major Issues: Key Experiments Required for Acceptance**

Reviewer #1: while the biological question of this study is intron splice variation... my main question is whether a potential technical issue with the use of cdhit may have influenced the interpretation of AS transcript heterogeneity. that is, whether post transcriptional mRNA sequence modifications that occur naturally, or perhaps triggered by infection, might be falsely counted as alternate transcripts based on altered sequence, but due to SNP changes rather than exon splicing/structural (indel?) changes. authors please comment on the potential for this based on their methods, and if it is a potential issue, address it accordingly by distinguishing SNP-differentiated from intron-differentiated alternate transcripts

I would have liked to see more detail (and a figure) describing how AS occurs in known (not candidates) effectors of Cfulvum, as I understand the results there are only a few affected by AS, but as several effectors in Cfulvum are well studied this should be quite interesting

i.e. what is the sequence pre/post AS, which version occurs during infection or non-infection, how functional motifs and known/predicted 3D structure is altered by inclusion/exclusion of the intron regions, etc...

Reviewer #2: The authors have done a great job examining the surprising levels of alternative splicing in relation to the functions of genes and their locations in the genome. My only major comment is that it would have been interesting to also see a figure where the frequency of alternative splicing is also examined in light of the physical characteristics of genes (e.g., their length, number of domains, GC content, etc.). Given that some of the events seem to be associated with transcriptional noise / heterogeneity, it may very well be that this noise is in turn associated with certain gene characteristics.

Reviewer #3: Here is my major concerns.

1. The study relies on the quality of transcriptomic assemblies. Although the package StringTie is considered as a good reference-based transcriptomic assembler and the authors employed a gene-by-gene assembly strategy to reduce the artifact, the assembly is probably still very noisy.

2. The study also did not consider the polymorphisms between the Race 4 and Race 5 genomes. Structural variation and other polymorphisms could significantly influence the results, which likely affects the conclusions drawn in the manuscript.

3. The manuscript could be improved if time-series expression patterns of AS isoforms could be described. It may identify genes showing interesting changing patterns over infection.

**Part III – Minor Issues: Editorial and Data Presentation Modifications**

Reviewer #1: the figure images as uploaded to the combined pdf are very low resolution and need to be improved

Reviewer #2: Line 3: “higher eukaryotes” - there are no “higher” and “lower” organisms; better say “plants and mammals”, which I believe is what you mean.

Line 64: "adaptions" -> "adaptations"

Line 373: “in contract” -> “in contrast”

Reviewer #3: Other comments:

1. “The transcript isoforms were then mapped to the genes of each isolate, and genes associated with more than one transcript isoform were now essentially considered as AS.”

How about partial assembled transcripts? For example, if a transcript is shorter than the full transcript, would the gene be considered to be an AS gene?

3. The dispensable chromosome Chr14 of Race 5 is absent in Race 4. How did the dispensable chromosome affect the liftover process?

4. When expression of transcripts was quantified, how did the quantification approach (SALMON) handle the common sequences between isoforms of a gene?

PLOS authors have the option to publish the peer review history of their article (what does this mean?). If published, this will include your full peer review and any attached files.

Reviewer #1: **Yes: **James Hane

Reviewer #2: No

Reviewer #3: No
---

## [Decision Letter · Decision Letter 1]

25 Nov 2024

Dear Prof. Dr Stergiopoulos,

We are pleased to inform you that your manuscript 'Transcriptome analysis of two isolates of the tomato pathogen Cladosporium fulvum, uncovers genome-wide patterns of alternative splicing during a host infection cycle.' has been provisionally accepted for publication in PLOS Pathogens.

Best regards,

Richard A Wilson

Academic Editor

PLOS Pathogens

Bart Thomma

Section Editor

PLOS Pathogens

Michael Malim

Editor-in-Chief

PLOS Pathogens

orcid.org/0000-0002-7699-2064

Reviewer Comments (if any, and for reference):

Reviewer's Responses to Questions

**Part I - Summary**

Reviewer #1: The authors have address my previous comments thoroughly and the manuscript is of a high standard.

Reviewer #2: The authors have done a good job addressing my comments / concerns. i have no further comments.

Reviewer #3: The authors made great efforts to address all comments. Although the revision did eliminate all concerns, reasonable tools and strategies were implemented to address biological questions of interest. I have no further comments.

**Part II – Major Issues: Key Experiments Required for Acceptance**

Reviewer #1: (No Response)

Reviewer #2: (No Response)

Reviewer #3: (No Response)

**Part III – Minor Issues: Editorial and Data Presentation Modifications**

Reviewer #1: the zenodo link in the data avialability statement should be adjusted to a DOI link

Reviewer #2: (No Response)

Reviewer #3: (No Response)

PLOS authors have the option to publish the peer review history of their article (what does this mean?). If published, this will include your full peer review and any attached files.

Reviewer #1: **Yes: **James Hane

Reviewer #2: No

Reviewer #3: No

---

## [Editor Report · Acceptance letter]

11 Dec 2024

Dear Prof. Dr Stergiopoulos,

We are delighted to inform you that your manuscript, "Transcriptome analysis of two isolates of the tomato pathogen Cladosporium fulvum, uncovers genome-wide patterns of alternative splicing during a host infection cycle.," has been formally accepted for publication in PLOS Pathogens.

Best regards,

Sumita Bhaduri-McIntosh

Editor-in-Chief

PLOS Pathogens

orcid.org/0000-0003-2946-9497

Michael Malim

Editor-in-Chief

PLOS Pathogens

orcid.org/0000-0002-7699-2064